# Track-dependency of tropical cyclone risk in South Korea

Chaehyeon C. Nam[1,*], Doo-Sun R. Park[1,**], Chang-Hoi Ho[1], Deliang Chen[2]

[1]School of Earth and Environmental Sciences, Seoul National University, Seoul, South Korea
[2]Department of Earth Sciences, University of Gothenburg, Gothenburg, Sweden

*Current affiliation: Department of Atmospheric Science, Colorado State University, Colorado, United States
**Current affiliation: Department of Earth Sciences, Chosun University, Gwangju, South Korea

*Correspondence to*: Doo-Sun R. Park (dsrpark@chosun.ac.kr)

**Abstract.** Several previous studies on tropical cyclone (TC) risk assessment have attempted to quantify the relationship between TC damage and its elements (i.e., exposure, vulnerability, and hazard). For hazard parameters, TC intensity (e.g. central minimum pressure, maximum wind speed) and size information (e.g. 30 knot radius of the TC) have been widely utilised. Our risk analysis of 85 TCs that made landfall in South Korea from 1979–2010, however, suggests that a small deviation of the TC track in the west-east direction ($\leq 250$ km, smaller than the average radius of TC) has a more dominant effect on the extent and distribution of TC damage than TC intensity/size. This significant track-dependency of TC damage exists because the TC track is responsible for the realisation of hazard change from potential to active. More specifically, although two TCs may have the same intensity and size, locally experienced rainfall and wind speed can vary according to their tracks due to topography. These results suggest that track information should be considered more carefully in assessments of future TC risk.

## 1 Introduction

Tropical cyclones (TCs) are among the biggest concerns for disaster management. As a single natural hazard worldwide, TCs are the costliest natural disaster (http://emdat.be). Many researchers have tried to understand and predict TC activity and associated risk. In TC risk studies, the risk triangle concept, which describes risk as comprising of three major elements (i.e. hazard, exposure, and vulnerability), is widely adopted (Mendelsohn et al. 2012; Peduzzi et al. 2012). In many empirical statistical models for TC risk, risk is estimated quantitatively using actual damage as a response variable (Pielke et al. 2008; Park et al. 2015). However, damage is more likely a materialisation of risk in the strictest sense (Cardona et al. 2012). The three risk elements are then used as explanatory variables in these studies. Exposure and vulnerability are usually expressed by the number of residents and regional gross domestic product (GDP) in the area of interest, respectively (Pielke et al. 2008). Hazard is typically represented by TC intensity parameters, such as central pressure and maximum wind speed (Nordhaus 2010; Hsiang and Narita 2012). Several recent studies have also suggested TC size as a hazard parameter (Czajkowski and Done 2014; Zhai and Jiang 2014).

Using TC intensity or size as hazard parameters, however, is insufficient for estimating TC damage. Even when a TC has the same intensity and size, damage can drastically change depending on its track, which causes the TC to experience different topography. Typhoon Rusa (2002) and Typhoon Haiyan (2013) are good examples. The record-breaking rainfall in Gangneung, South Korea was incurred by Typhoon Rusa (2002) because the track of Typhoon Rusa (2002) was optimal for the orographic lifting effect, and so heavy precipitation occurred over the city (Park and Lee 2007). The deadliest damage by typhoon Haiyan (2013) in the Philippines primarily came about because the TC penetrated Tacloban city, which is located in a low-lying area near the ocean, such that most of the damage arose from storm surge (Ching et al. 2015). In both cases, if the TCs went through a different area, avoiding the mountains and lowland, the result could have been much less devastating.

This study focused on the role of track in the TC risk determination process. We tried to directly compare the priority among the risk elements by using various statistical analyses of the historical TC records and damage data from South Korea and to explicitly show the significance of track-dependency in TC risk. Moreover, we explained the role of track within the TC risk triangle framework and how TCs with similar intensity and size but slightly different track patterns could bring dramatically different risk patterns. The rest of the paper is organised as follows. Section 2 lists the data sets for TC intensity/size, local wind and rain hazard, damage, and social index used in this study, and explains how these datasets were processed and statistically analysed. Results from the risk comparison and decision tree analysis are described in Sect. 3 and a summary of the major findings of this study is given in Sect. 4. Finally, in Sect. 5, we address several implications for future risk research and predictions based on our findings.

## 2 Data and Methods

### 2.1 Data Source and Processing

The present study utilised several data sets: 1) weather station data, 2) TC track, intensity, and size data, 3) national survey data of TC damage, and 4) national survey data of regional wealth. This section will describe how different data sets were obtained and processed before the statistical risk analysis.

First, from 60 weather stations throughout South Korea (see Fig. 4 for station locations), daily maximum near-surface wind speed and daily accumulated precipitation values were gathered.

Second, TC information including track, intensity, and size was obtained from the Regional Specialized Meteorological Center (RSMC) best-track data. For intensity, we used the maximum wind speed and central pressure data. For TC size, we used the largest radius of 30 knot winds, which is specifically provided by RSMC. RSMC best-track data in the 6-hour interval was interpolated to a 1-hour interval to obtain precise hazard values at landfall (Park et al. 2011). The interpolated RSMC best-track data was used to select the TCs that made landfall to South Korea (TCs entering the area within 3° of the coastline of Korea). Then, we verified these TCs with the official influential TC record in the Typhoon White Book issued by the Korean National Typhoon Center (NTC, 2011), as in our previous study (refer to Park et al. (2016) for more details on NTC Typhoon White Book). We chose only the TCs that had maximum wind speeds greater than or equal to 17 m s$^{-1}$ (above or equal to Tropical Storm (TS) class) at the time of

entering the 3° line. As a result, 85 TCs were determined to be influential in South Korea from 1979 to 2010 (see Supplementary Table 3 for the list of the names and years of all the target TCs).

Next, we used damage data from the National Disaster Information Center (NDIC) of the Korean government (http://www.safekorea.go.kr), after following procedures. NDIC property loss data consists of data on monetary damages to industrial, public, and private facilities, standardized to the value of money in 2005 by accounting for inflation. The loss data were collected by local governmental offices and therefore most losses could be reported regardless of whether the victims were insured or uninsured. There may, however, be some case of minor losses that were not reported to the local offices by the victims. The raw dataset included damage data caused by all types of extreme weather such as TCs, heavy monsoon rainfall, heavy snowfall, or high waves. Some cases were not classified by specific damage sources. Some cases were categorised under high-wave damage although they were also caused by TCs as the high waves were induced by TCs. Therefore, we matched all the raw loss data to the TC data using the NTC White Book (NTC 2011), RSMC best track data, and the time periods. To be more specific, we compared the three relevant periods: 1) the period of warning issued recorded in the NTC White Book (NTC 2011), 2) the number of days that TC stayed within 3° of the Korean coastline based on the RSMC dataset, and 3) the period of damage occurrence recorded in the NDIC dataset[1]. If any day of the NDIC damage period overlapped with the RSMC or White Book influence period of a TC, the loss was attributed to the TC. "No damage" and "Damage" cases were later categorised based on whether there existed any economic loss records reported by NDIC for the given province and TC event.

Third, Province-level aggregated wealth data was obtained from government statistical surveys (Korean Statistical Information Service, http://kosis.kr/). We aggregated the 17 districts of South Korea into 5 provinces, because the administrative division had been changed between 1979 and 2010, and the size of the 17 districts varies from city-size to province-size. The names of the provinces are Gyeong-gi (GG), Chung-cheong (CC), Jolla (JL), Gang-won (GW), and Gyeong-sang (GS) (See Fig. 2 for the distribution of the provinces). These five provinces have independent records of damage for every influential TC case and annual regional wealth. The temporal variation of wealth was considered through the normalization of damage data to the reference year, 2005, with wealth per capita. In general, the wealth of South Korea has consistently increased. However, there are significant differences in the growth rates among provinces, which affect the TC damage records. Through normalization, the potential impact of regional differences in wealth trends was eliminated. The spatial disparity of wealth at a certain time (i.e. 2005) should be addressed when mapping the damage distribution.

---

[1] NDIC cannot differentiate the damage from multiple hazards when there are multiple successive extreme phenomena. For example, if heavy rainfall watch started on July 15th and then a TC came to South Korea on July 20th and decayed on July 22th and there was no gap between the rainfall and TC advisories, NDIC aggregates the damage amounts and record the damage period as July 15th to 22th. Therefore, to confine the origin of the loss data to one TC, we excluded cases whose damage period exceed five days from landfall.

## 2.2 Statistical Analysis

### 2.2.1 Data mining methods

The 85-selected influential TCs were then grouped according to their track patterns using the fuzzy c-means clustering method (FCM). We clustered the track patterns, considering only the parts of the tracks in the domain of 28° N–40° N and 120° E–138° E (grey boxes in Fig. 2) so that we could divide tracks and focus on the paths near South Korea. There may be other ways to categorize track pattern in the area of interest. For example, one can group them using a certain longitude criterion (e.g. east versus west from 128 E) or the approaching angle criterion (Hall and Sobel, 2013). Here, we chose to use the FCM, as it is widely adopted for objectively dividing widespread data with amorphous boundaries. Some previous studies have shown this method to be effective for grouping TC track patterns (Kim et al. 2011; Park et al. 2017).

The TCs were grouped into four types. The optimum cluster numbers were decided using four validity measures: partition coefficient, partition index, separation index (i.e. Xie and Beni index), and Dunn index. The partition coefficient measures how much overlapping the fuzzy clusters have, and inversely proportional to the average overlap between the clusters. Both of the partition and separation indexes are computed by compactness and separation of the clusters. However, the partition index represents separation as the sum of the distances between the clusters while the separation index does as the minimum of them. The Dunn index is calculated by the ratio of the shortest and the longest distances of the two objectives within a same cluster. The larger partition coefficient and smaller partition index, separation index, and Dunn index create better clustering (for a more detailed explanation and formula of validity measures for the optimum cluster number, refer to Appendix B of Kim et al. 2011). All the indexes pointed to four being the optimum number in our case. We conducted some sensitivity tests that introduced slight changes to the TC lists, such as different time frame (e.g. 1979–2015) or different clustering domain (e.g. 5° area from the Korean Peninsula coastline), and four still appeared to be the optimum cluster number from the validity measures.

We further introduced the decision tree analysis to decipher the relationships among risk elements. The decision tree method, a multi-variable technique, allowed us to explain, describe, classify, and predict a target as a result of the combined effects of multiple input variables beyond a one-cause and one-effect relationship. Compared to other multi-variable techniques, the decision tree method's advantage is that it is easy to use, robust with a variety of data, and most of all, intuitively interpretable. It helps decision analysts to structure the decision process in a graphical sequence.

Among several famous decision tree algorithms, this study applied See5/C5.0 as a classification method for TC risk materialisation. The See5/C5.0 algorithm is an improved version of C4.5 (Quinlan 1993) in terms of accuracy, speed, and computer memory consumption. Further, the C4.5 algorithm was advantageous because it could accommodate all the required class, binary, and continuous variable types (see Supplementary Table 1). See5/C5.0 calculates the information gained at each node, based on the entropy concept, in order to select the most efficient attribute for splitting the training samples into two branches.

To prevent over-fitting, we introduced pruning and cross-validation. First, we required that branches have a sample size of at least five. The number five was determined through the retrospective pruning process.

Second, a ten-fold cross-validation, which divided the training data set and validation data set randomly ten times, was conducted. Cross-validation results are provided in Supplementary Table 4, and they show the decision tree results (e.g. model accuracy, tree size, or attribute usage) are stable and consistent. The best-track data based decision tree has a relatively broad range of distribution in terms of size and accuracy for each training data set, but the significant track-dependency remained through the cross-validation.

### 2.2.2 Significance tests

For all the statistical analysis of risk comparison among track groups, nonparametric methods were used (Sawilowsky 1990). Medians were used rather than means, and rank-based procedures were conducted for any significance test. This is because we cannot regard the TC damage as following a normal distribution; rather, damage shows an extreme distribution. Zero losses were recorded for 30% of TCs, and 30% of all the accumulated damages were attributed to a single TC, Typhoon Rusa in 2012. The Kruskal-Wallis test, or the one-way Analysis of Variance (ANOVA) on ranks, was used to determine if there are statistically significant differences for a variable between track-groups. Spearman's rank correlation coefficient, which measures the linear relationships between the rankings of two variables, was used instead of the more common Pearson product-moment correlation coefficient, which measures linear relationships between the raw values of two variables.

### 3. Results

In this paper, we adopted the hazard mode concept (**potential** versus **active hazard**) from the risk management field (MacCollum 2006). For the hazard mode concept, active hazard refers to a situation when "a harmful incident involving the hazard has actually occurred", whereas potential hazard refers to a situation where "the environment is currently affected but not yet activated at a given place and time". By this definition, we refer to heavy rainfall and wind gust induced at the local area by the TC as active hazards, and we consider the TC system's minimum central pressure, maximum wind speed, and size over South Korea as potential hazards. These two modes of TC hazard (potential and active) are utilised throughout this paper.

### 3.1 TC hazards and risk of different track types

In order to objectively evaluate the effect of each TC track on damage, a total of 85 TCs which influenced South Korea during 1979–2010 were grouped into four track patterns. The four TC track patterns can be characterised as 1) east-short, 2) east-long, 3) west-long, and 4) west-short types based on the position and length of the TC tracks around the Korean Peninsula (Fig. 2). Although the average zonal distance between the mean tracks of east-types (i.e. east-short and east-long) and west-types (i.e. west-short and west-long) was only about 250 km, hazards (both potential and active) and damages caused by the TCs are significantly different depending on the four TC track patterns at the 99% confidence level based on the Kruskal-Wallis test (Fig. 3). This highlights the importance of track in TC risk assessment because the 250 km distance is not long considering that the average errors of track forecasting in the western North Pacific are about 200 and 400 km for 24 and 48 hours, respectively (Roy and Kovordanyi 2012).

Meanwhile, the high sensitivity of damage on the track shown in Fig. 3 suggests that the current skill of TC track forecasting may not be enough to exactly estimate TC risk distribution over South Korea in advance of 1 day and over.

As shown in Fig. 3, potential hazards display different results from active ones, although both originate from the same TC. Potential hazards are stronger in longer tracks, i.e. east-long and west-long, while active ones are stronger in west-types than east-types (compare Figs. 3(a)–(c) to Figs. 3(d)–(f)). In addition, even if potential hazard parameters have been widely used in TC risk analysis (Nordhaus 2010; Hsiang and Narita 2012; Czajkowski and Done 2014; Zhai and Jiang 2014), they show worse accordance with damage than active ones. For all potential hazards, the ranking is in order of east-long, west-long, west-short, and east-short. It is natural for a TC with longer track to have higher wind speed, deeper central pressure, and larger size, since a TC with a stronger intensity should be more durable compared to a weaker TC under the same environmental conditions, such as friction, vertical wind shear, and sea surface temperature (Kim et al. 2011). In contrast, for active hazards, the ranking is in order of west-types, east-long, and east-short track patterns. This relationship between track and active hazard parameters (i.e. near-surface wind, rainfall, and influence duration) cannot be simply explained unlike potential hazard parameters.

Figure 4 shows the spatial distribution of active hazard parameters with topography for each track pattern. Looking at the near-surface winds (Figs. 4(a)–(d)), the near-surface winds of west-type tracks were comparable to near-surface wind of east-long track, particularly along the coast, even though potential hazards of west-type tracks are significantly weaker than those of east-long tracks (Figs. 3(a)–(f)). This can be attributed to the concepts of dangerous and navigable semicircles. In the case of west-type tracks, South Korea falls within a dangerous semicircle (right-hand side of the direction of TC movement), in which the TC translation speed and rotational wind field are additive, and hence strong wind speed is observed therein. In contrast, in the case of east-type tracks, the country is located under a navigable semicircle (left-hand side of the direction of TC movement), in which the TC translation is counter-directional to the rotational wind. Therefore, weaker wind speeds are found there than that in the dangerous semicircle.

In terms of rainfall, much heavier rainfall was found in the west-type tracks than that in east-type tracks along mountainous area, particularly the Sobaek mountains (Figs. 2(c) and 4(e)–(h)). Heavy rainfall along the mountains can be explained by the orographic lifting effect. When a TC is located in the southwest of South Korea, the eastern sides of the Sobaek mountains become the upstream slope of the tangential wind of the TC, causing more torrential rainfall than the inherent rain band of the TC (Park and Lee 2007; Lin et al. 2002). Thus, the orographic lifting effect can be maximised by west-type tracks but not east-type ones.

Finally, the influence duration was distinctly longer for west-type tracks compared to east-type tracks. Only a TC with a west-long track penetrates the country, and hence the west-approaching TCs could affect a more extensive area for a longer time. The long influence duration of the west-short track was possibly because a TC with a west-short track moves the slowest, so that it could stay for the longest time (significant at the 95% confidence level, Kruskal-Wallis test). Note that we calculated influence duration for each station by applying the same criteria for wind and rainfall. A station was marked as "influenced" if either the daily accumulated precipitation or daily maximum sustained wind speed recorded at that

station on the specific day exceeded the station's critical thresholds, which we set as the 90$^{th}$ percentile of each station.

Looking at Fig. 3, the ranking of active hazards was exactly same as that of damage. In addition, the spatial distribution of damage also matched well with those of active hazards (compare Figs. 4 and 5). The area where active hazards are high, exhibited high risks. There was only one exception, for west-types, the southwestern province (Jeonla, JL) recorded less damage than the southeastern province (Gyeongsang, GS), although stronger active hazards are found in JL than GS. This discordance was partly explained by exposure disparity. GS possesses higher wealth compared to JL. After the damage was divided by regional wealth (parentheses of Fig. 5), the spatial distribution of damage became more analogous to that of active hazards, even if the damage in GS is still slightly higher than in JL for the west-short tracks (Fig. 5(d)). This may be related to different vulnerability to TCs between the two provinces. Since GS is more mountainous than JL, the vulnerability of GS to TC rainfall can be higher than that of JL. Nevertheless, all the active hazard parameters (r=0.62) showed much higher correlations with damages than potential hazard parameters (r=0.29), although most of the potential hazards displayed statistically significant correlations at the 95% or 99% confidence levels (Table 1).

All the results in this section suggest that active hazards are better indicators of TC risk than potential ones. In other words, if we want to a predict whether there would be damage to a city or not and if the active hazard information is available, we may not need to gain any additional information of potential hazards. Nevertheless, this does not mean that potential hazards are not important. We have to utilise a climate model with a fine resolution of at least less than 10 km for realistic simulation of active hazards, i.e. wind and rainfall (Park and Lee 2007; Lee and Choi 2010), which is a difficult task and requires high computing power. Hence, active hazards seem not to be optimal for risk forecasting to help emergent decision making, as well as climate change research with large spatiotemporal scale. In this respect, it can be more valuable to use potential hazards for risk assessment if we identify an additional factor that can fill the gap between active and potential hazards.

## 3.2 Importance of track in TC risk analysis

Here, through the decision tree analysis, the importance of track in TC risk was investigated. Through the decision tree analysis, the following three questions could be answered: 1) what is the most effective factor for classifying "damage" and "no damage" cases for TCs making landfall to South Korea, 2) how do different factors in combination determine damage occurrence, and 3) what critical values of the factors can be used as quantitative guidelines related to TC damage occurrence? Here, the decision tree model was designed to objectively classify whether a TC will bring damage to a province or not; the decision tree used potential hazards and track as input variables (see Supplementary Table 1 for more information about the input variables). Overall, we had 355 effective cases, comprising 160 "damage" cases and 195 "no damage" cases; we only considered damage occurrence in each province by a TC (see Supplementary Table 2 for detailed information of damage cases).

According to the decision tree, track information acts as the primary determinant of TC risk. Information about track pattern was nominated as the first splitting attribute (Fig. 6). This means all 355 cases should be classified by track group prior to all other decision nodes, in order to reach the end nodes. In other words, the most important factors of TC risk may be neglected when performing a risk analysis without

track information. The detailed process is as follows. First, the model simply sent all west-type TCs to the end node of "damage". Next, the east-type TC cases were assessed according to province and TC intensity (maximum wind speed). For a TC in the east-long group, damage could occur in JL, GS, and GW provinces. Particularly, for the GW province, damage will be inflicted only if the maximum wind speed is greater than 41.1 m s$^{-1}$. East-short cases, unlike east-long cases, were sent to the intensity criterion before the province criterion. East-short TCs with weak intensity (maximum wind speed less than 25.7 m s$^{-1}$) were directly linked to the "no damage" node. The east-short TCs with satisfactory intensity (maximum wind speed greater than 25.7 m s$^{-1}$) were sent to the province criterion; even though it is a small portion, the critically strong east-short TCs can incur damages in the southern provinces.

The relative importance of the variables in each decision tree was offered quantitatively in terms of the usage rate by the See5/C5.0 algorithm. When an attribute is the most-related variable to the target variable, the attribute should be used most frequently for classification by a decision tree model. In our decision tree (Fig. 6), the track group variable was used 100% of the cases; province and potential hazards then follow with usage rates of 48% and 37%, respectively. Therefore, we can say that for risk determination, TC track was the most important attribute, which gives essential information on TC risk analysis. The use of the province variable as the second most important variable was mainly related to the relative location of the province with respect to TC centre along the track. Southern provinces are generally closer to the TC centre regardless of the four track types because TCs move from the south (low-latitude) to the north (high-latitude). Potential hazards were the third most important attribute. Maximum wind speed was utilised as an effective classifier, but TC size was not used.

## 4 Conclusion and discussion

Our results show that potential hazards, generally utilised in risk analysis, are less correlated with damage than active hazards. However, potential hazards are still valuable in risk analysis considering their convenience. In addition, according to our analysis, track information can considerably fill the gap between potential (e.g. maximum wind speed, and central minimum pressure) and active hazards (e.g. near-surface wind speed and rainfall). Figure 1 shows the graphical model summarising the above points, which indicates the position of the track in causality relationship to the TC risk process. Track may contribute to realising active hazards through altering the following factors: 1) interaction with inhomogeneous topography, 2) storm-relative location (i.e. which quadrangle of TC the city is located in), and 3) influence duration. The decision tree analysis suggests TC track as the most decisive factor for TC damage occurrence, whereas potential hazards play only peripheral roles. Therefore, it is recommended to utilise track information as an additional factor when using potential hazards in risk analysis. Our results also imply that it is necessary to consider possible large uncertainty in future TC risk projection because of high sensitivity of TC risk on track, as well as the lack of reliability of future projection of TC tracks (Knutson et al. 2010, Walsh et al. 2015).

On the other hand, the importance of track may differ by country because topography among the three factors suggested is not identical between countries. If a country has major mountainous area like South Korea, track information may become more important, and vice versa. The dependence of track in TC risk over Southeastern United States, for example, in which there is little mountainous area, may be less

important than that of South Korea. As a future study, we would compare role of track in TC risk between countries having different topographic conditions.

Our conclusion not only highlights the importance of track in TC risk analysis, but also suggests that track pattern type can be used as an independent variable for regional risk forecast. However, the decision tree model utilised here is not proper for forecasting, since it is prone to overfitting and errors due to bias and variance. This is because the decision tree determines an optimal choice at each node. Choosing the best answer at each step does not guarantee the global optimum. If the model makes a different choice at a given step, the final node can be totally different, especially when the dataset is small. For the current study, to prevent these errors, we verified our results with pruning and cross-validation. We also used the decision tree method to diagnose the relationship between risk elements but not for forecasting. As a further study, we plan to utilise the random forest model for forecasting.

**Acknowledgement**

This work was supported by the Korea Ministry of Environment (MOE) as "Climate Change Correspondence Program" and the Korea Meteorological Administration Research and Development Program under Grant KMI [2018-03413].

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

**Table 1: Each Spearman's correlation coefficient of property losses with active and TC-based hazards.** Active hazards are maximum daily wind speed, maximum daily precipitation, and the sum of influenced periods for all 60 weather stations. TC-based, potential hazards are maximum wind speed, central pressure, and storm radius (30 knots) based on the RSMC best-track data for each track group. The significances of correlations are shown with asterisks.

| | **Four track groups** | | | | **All** |
|---|---|---|---|---|---|
| | East-short | East-long | West-long | West-short | |
| **Active hazard parameters (from weather station)** | | | | | |
| Daily max wind speed | 0.45** | 0.58** | 0.66** | 0.59** | 0.62** |
| Daily precipitation | 0.37** | 0.66** | 0.74** | 0.80** | 0.71** |
| Influence duration | 0.48** | 0.76** | 0.59** | 0.78** | 0.76** |
| **Potential hazard parameters (from best-track data)** | | | | | |
| Maximum wind speed | 0.39** | 0.17* | 0.27* | 0.39** | 0.29** |
| Central pressure | -0.40** | -0.16* | -0.35** | -0.41** | -0.27** |
| Storm radius | 0.39** | 0.08 | 0.16 | 0.30* | 0.24** |

5    * Significant at the 95%, ** significant at the 99% confidence levels.

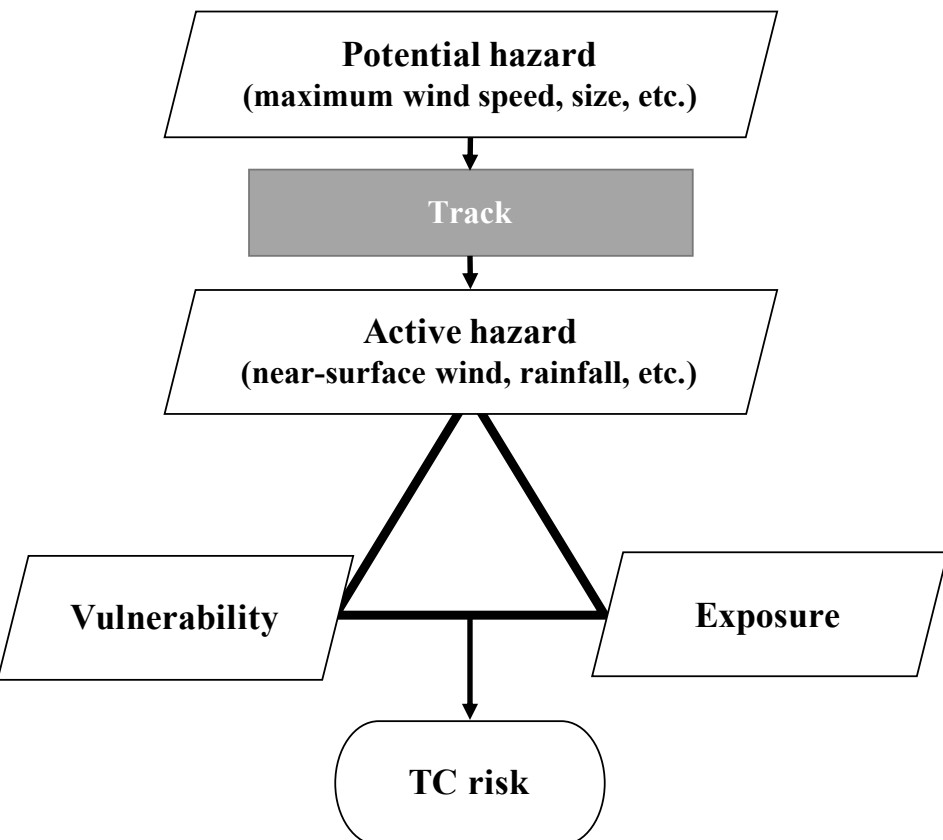

**Figure 1: Flowchart for local risk materialisation process with TC risk elements and their relationships.** Potential and active hazards correspond to indirect and direct causes for TC risk in terms of causality science. See the main text for more explanation.

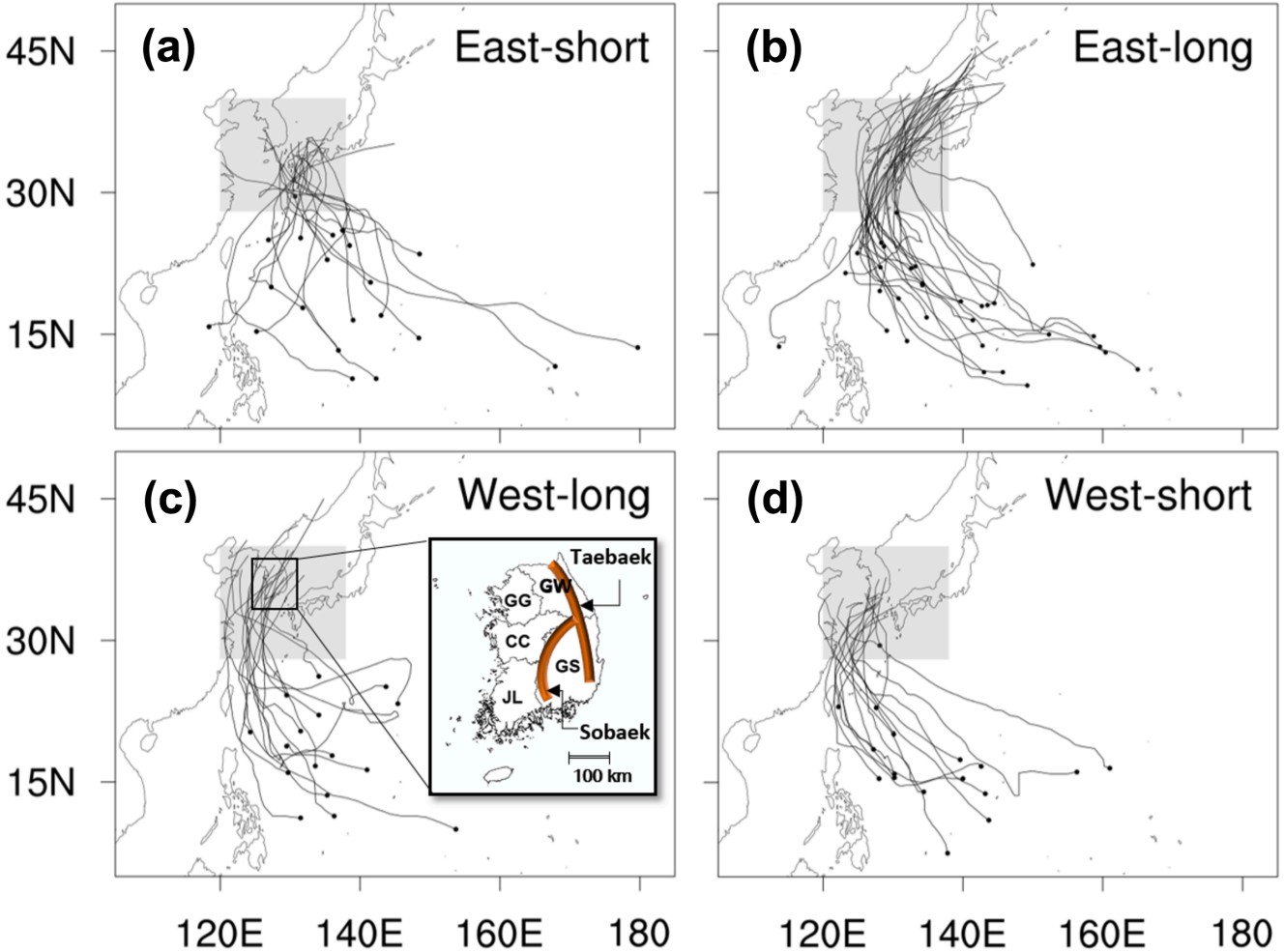

**Figure 2: Four groups of tropical cyclone tracks that made landfall over South Korea from 1979–2010.** The box shaded in grey, covering 28 N–40 N and 120 E–138 E, indicates the clustering domain for the fuzzy c-means clustering method. A map of the five aggregated provinces of South Korea is displayed in **(c)**: Gyeong-gi (GG), Chung-cheong (CC), Jolla (JL), Gang-won (GW), and Gyeong-sang (GS). Taebaek and Sobaek mountains are indicated with orange lines.

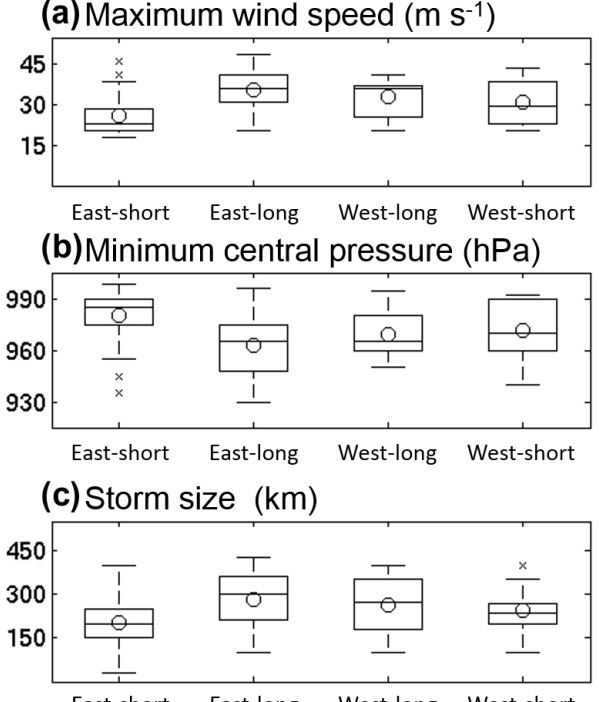

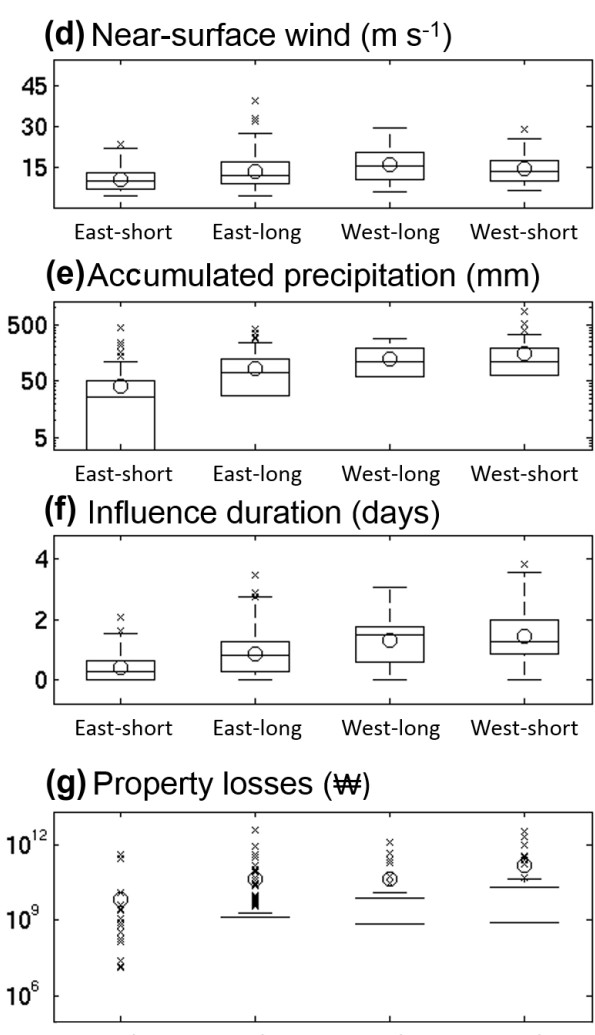

**Figure 3: Boxplots of the hazards and damages of track-pattern groups. (a)** Maximum wind speed, **(b)** central minimum pressure, and **(c)** storm size from RSMC best-track data at the point that the TCs entered the area within 3° of the coast of the Korean Peninsula, or for the TCs that did not enter the area within 3°, when they were closest to South Korea. **(d)** Daily maximum wind speed (10 min average), **(e)** daily accumulated precipitation, **(f)** influence duration from 60 weather stations, and **(g)** property losses (1$ ≒ 10³₩) over South Korea. The storm size is the longest radius of 30 knot winds or greater. Station maximum wind speed and precipitation are one maximum daily value in the whole influence duration. The upper (bottom) side of each box is 0.75 (0.25) quantile. The bar inside the box represents the median, and the circle represents the mean. The plotted whiskers extend to most extreme data point that is not an outlier. Outliers, which are located outside of the maximum whisker length, are drawn as 'x.' The maximum whisker length is 1.5 times the value of the third quantile minus the first quantile. Note that y-axes of (e) and (g) are in log scale, and zero cannot be shown in these plots (i.e. the third quantile (0.75) of east-short in (e), the third quantiles of all four track-types in (g) and medians of east types, even the first quantile for east-short type in (g)).

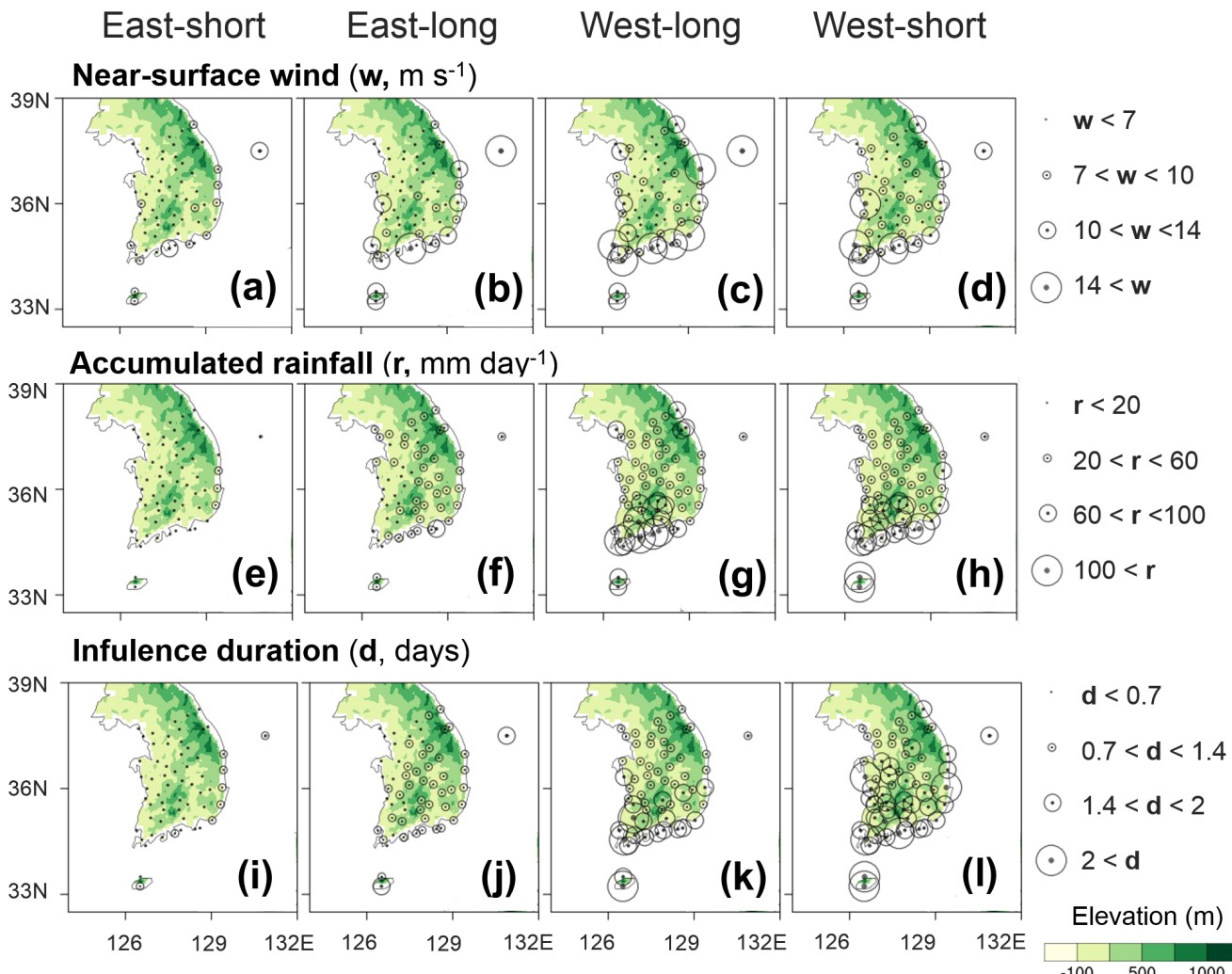

**Figure 4: Three active hazard parameters – wind, precipitation, and duration, of tropical cyclones for each track-type observed at 60 weather stations.** Shown here is the mean value recorded at each station. For example, for a certain station, if a TC recorded above-threshold value for the station for three days (refer to the method section for the definition of a threshold), influence duration is '3' for that TC at that station. Then, we get 22 influence duration values for east-short type, because there are 22 TCs of east-short type. We take the mean of the 22 influence duration values and plot it with different sizes of circles at the locations of the stations, respectively. In addition to duration, for wind and rain, we used daily maximum wind speed (10 min mean) and daily accumulated rainfall. When we have multiple days of influence duration, we have multiple values of daily maximum wind speed and daily accumulated rainfall for the specific TC. We then used the biggest wind/rain value among the multiple values as the representative value of the hazard caused by that TC at that location.

## Medians of Regional Property Losses* for Each Track Cluster (1979-2010)

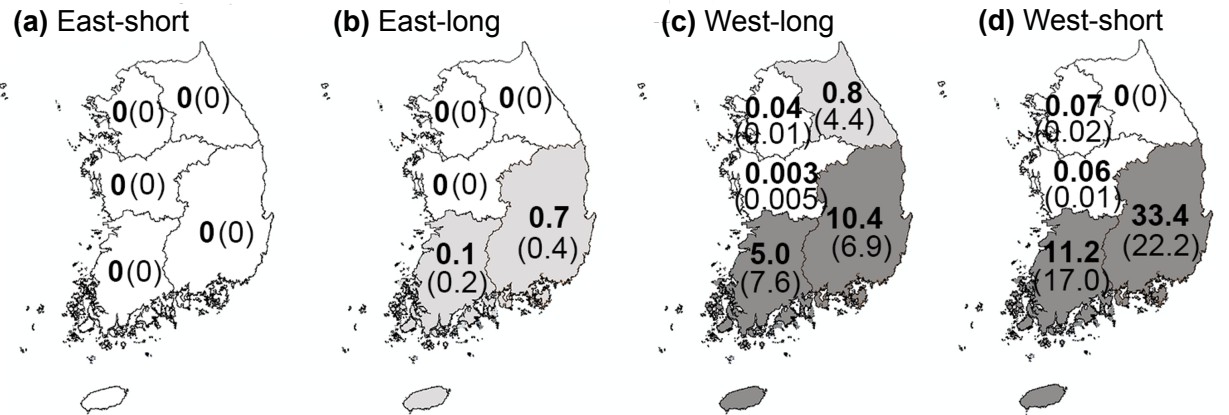

**(a) East-short**

**(b) East-long**

**(c) West-long**

**(d) West-short**

*Unit: **billion KRW** for the amount of property losses
(in parentheses, the ratio of property loss to the amount of regional wealth is shown in the unit of $10^{-6}$)

**Figure 5: Medians of regional economic losses from a tropical cyclone (regional economic losses divided by regional wealth).** The dark shading indicates provinces that have median losses larger than ₩(KRW) one billion, and the light shading indicates provinces with median losses larger than ₩ 0.1 billion and smaller than ₩ one billion. More than the half of the east-short TCs are undamaging TCs, so the property loss medians of all provinces are zero.

# TC best-track based decision tree for South Korea's 5 provinces

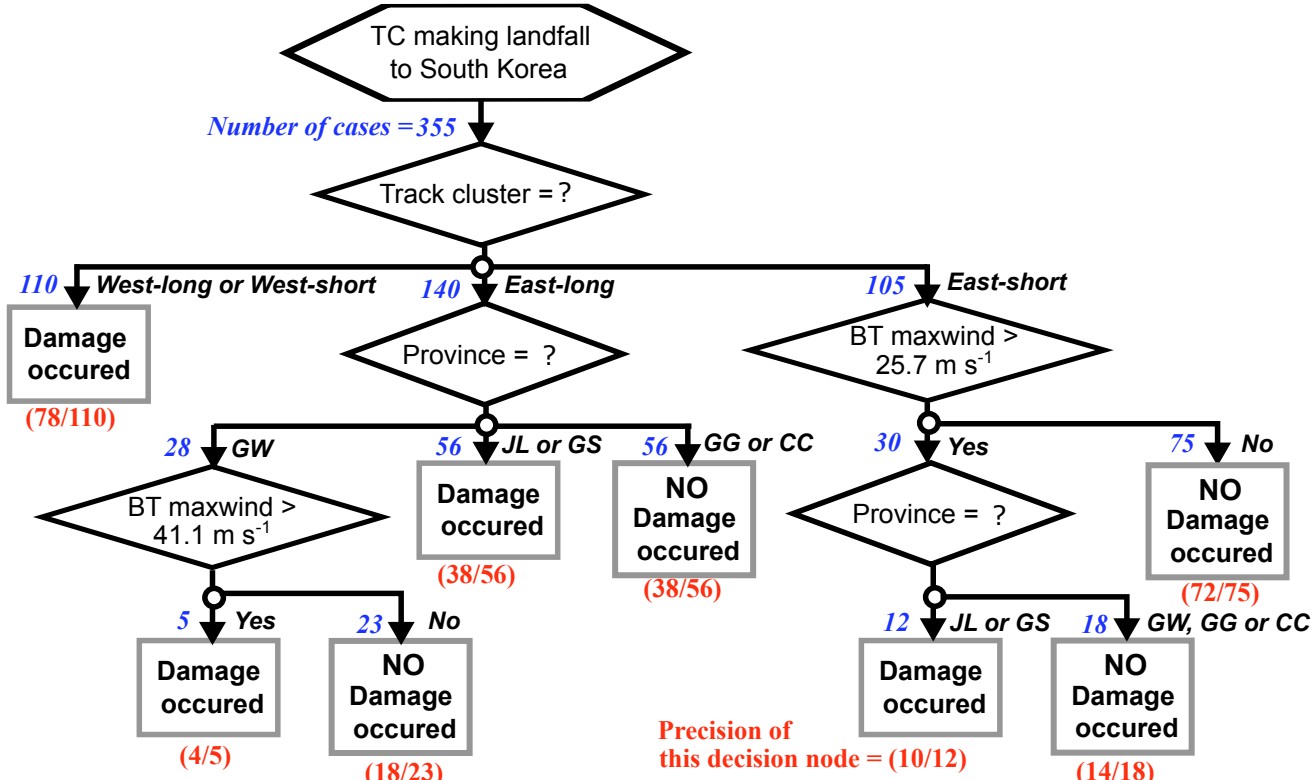

**Figure 6: Decision tree model for damage occurrence using the four TC best-track attributes (maximum wind speed, central pressure, storm size, and track-group) and province information as input variables**. The hexagonal box indicates the start of the algorithm, and the rhombus boxes contain questions bifurcating each node. The grey rectangular boxes indicate the final diagnosis boxes, in which the precision of the diagnosis is written in parentheses with red ink (the number of correctly identified cases / the number of cases diagnosed following the specific sequence of criteria). The number of cases corresponding to each criterion is presented at the left side of each arrow with blue ink, and at the right side of each arrow, there is corresponding answer for the question right above the rhombus box. Refer to Fig. 2 for the full names of provinces for each abbreviation (i.e. GW, JL, GS, GG and CC).