# Peer review of "Track-dependency of tropical cyclone risk in South Korea"

_Natural Hazards and Earth System Sciences, 2017_

## Referee Comment (RC1) · Anonymous Referee #1 · 31 Jan 2018

The article entitled "Track-dependency of tropical cyclone risk in South Korea" by Nam et al. addresses the issue of including cyclone tracks as the bridging factor between exposure and actual impacts of tropical cyclones. The article introduces the main problematic in the first section explaining the distinction between "active" hazards (e.g. rainfall and wind surges) and "potential" hazards (tropical cyclone intensity). Then, an extensive presentation of the datasets, tools and methods that the authors used is followed by a discussion of the results after applying a tree decision method that the authors propose for evaluating cyclones hazard.

The method introduced in this paper seems to provide meaningful results when applied to the Korean peninsula and I would recommend the article to be published in NHESS.

My only major concern goes on the presentation and organisation of the manuscript.

In several parts, I found that the text is repetitive on the arguments and methods and thus less attractive to the reader. Furthermore, my opinion is that the results section is too long and difficult to follow. It should be divided to subsections in order to ease the reader with better articulation of the main findings. Finally, although understandable language could be improved.

I would also recommend to the authors to include a discussion on the potential application of their method to different regions. It seems that the number of track clusters is detrimental for the complexity of the decision tree. Is the applicability of their method jeopardized if e.g. a broader region with complex geographical and exposure issues is taken into consideration?

Finally, I would recommend a demonstration of their method application for the ensemble members of a tropical cyclone forecast. What is the hazard variability when the spread of the forecasted tracks is relatively large?

---

## Editor Comment (EC1) · P. Lionello (Editor) · 14 Mar 2018

This paper shows that for tropical cyclones affecting South Korea, track is the main factor responsible for damages, being more important in a decision tree analysis than other factors such maximum wind speed or minimum atmospheric pressure level.

However, the manuscript needs to be extensively revised for being published. English (though understandable) needs to be improved, many sentences to be rephrased, several paragraph, in particular of the "Results" section to be better focused, figure captions to be more explicative, many typos to be corrected. Further the "Summary and conclusions" section does not deliver the main outcomes of the paper in a clear, direct and concise way. May be better to split it in two sections: "Discussion" and "Conclusions"

Moreover, it should be discussed whether this strong sensitivity of the risk on the tracks is a general characteristic of tropical cyclones or a peculiarity of the South Korea territory, depending on its characteristics in terms of morphology, land use and exposure. Tropical cyclones in the Gulf of Mexico hitting the southern United States would exhibit a similar dependence on their track position?

The difference between the tracks of the (west and east) cyclone clusters is about 250km. Is this difference sufficiently large to be predictable for an individual cyclone in advance? How many days before reaching South Korea? Authors mention that uncertainty in track of future projections should, therefore, be accounted for. Is this distance among clusters larger or smaller than uncertainty of projections? In there any indication of such a change for South Korea?

The supplement material contains three tables and a figure to support the description of the decision tree. However, the methodology for construction and validation of the decision tree is not described. I suggest to add a very short text describing it in the main body of the manuscript (in section "Material and methods") and use the supplement for providing more information.

Here is a list of minor comments (which is no way meant to be exhaustive)
Line 6, delete "of"
Line 7 Rephrase the sentence.
Line8 comma missing before "while". Delete "mainly"
Line 10 I suggest "to predict damage." or "to predict the occurrence of damage."
Insert "≤ 250 km" among brackets
Line 12 I suggest to replace "the other hazards developing from a potential to an active hazard" with "an hazard developing from a potential to an active one" …. But may be I do not understand the sentence. Please rephrase
Line 14-15 "physical geography experience, duration of influence, and relative position of dangerous semicircle side of the TC", this phrasing looks strange to me
Line 16, add comma after "modeling"
"risk modeling" or "risk assessment"?
What is meant with "trivial"?
Line 16 authors, apparently , in the final sentence of the abstract consider "error " and "uncertainty" as equivalent terms. In general they are not. Please explain better.

End of section 1: a description the content of the paper is missing
Section 2 replace "Materials and method" with "Data and methods"

Page 3 line 1, definition of TC size is unclear
Line 4 follwing, consist,
Line 5: "standardized to the value of money in 2005 and taking inflation into account" Should "and" be replace with "by"? otherwise it looks a duplication of the same concept
Line 6 avoid repeating "collected"
Line 8 was including
Line 10 bad phrasing, The damage was likely indeed caused by waves, which in turn were caused by the TC
Line 16 bad phrasing .I suggest replacing "if there exist any" with "the presence of"

Probably "if" should be replaced with "whether"
"Line 11 what is the definition of "influence period"?
Line 14 what is the "damage period"?
Line 25 delete "of"
Line 26 delete "of provinces"
Line 29 what is the "influence duration"? delete "also"
Line 31 rephrase "the range of duration was limited by the summation". Sentence is not clear

Line 1 "defined and distinguished" better "identified"?
Line 4-5 "influential" means having great influence on someone or something. This criterion does not account in any way for the size of the impact of the cyclone
Line 14 "more comprehensive" than what?
Line 18 replace "from" with "in"
"their intensity was above TS" replace with "their wind speed was above the TS threshold"
Line 29-31 The explanation of the grouping criteria and why four clusters have been used  is not clear to me.  What is here a validity measure? The  definition of the used indices is missing
Line 26 replace "not for the whole tracks from genesis to disappearance, but for" with "considering only"
Line 26-28 long sentence

*I stop here with comments on the English form*

Line 30 "Xie and Beni index, and Dunn index" should be briefly described

Line 16-17 what is meant here with stability and consistency of results? How was this checked?

Section RESULTS
line 1 distance between types? How is defined  the distance between two types
Line 11 "TC-based hazard ranking" on which variable is based? How is the ranking computed?
Line 15 … apparently a self-contradicting statement
line 11-12 sentence:  ranking based on which quantity?
In my view  the second paragraph not well focused. It addresses some differences between hazards rankings among clusters, between this and previous studies, and among hazards …all together . please explain better
I think that "damaged " or "undamaged" can refer to the territorial units, to the population, to the exposed goods, but not to a "case". The expression "damaged case" sounds odd to me
line 2 "navigable"?
Third paragraph long and not well focused
Line 29-30 repetition

Section CONCLUSION
In the text I cannot find a precise definition of Active hazard (it should be provided). Anyway, my  understanding is that they are hazard that actually produce  some losses, victims, accidents emergency. In such case the sentence at line 5-6 page 9 is trivial as it follows for the definition of active hazard .Pease explain better or deleted it.
Line 11 "When local active hazard information is missing, TC track acts to bridge the information gap between the TC system and local risk" is not clear to me

FIGURES
figure 2
first and third quantile values are   0.25 and 0.75? please be specific
The sentence "The plotted whiskers extend to the adjacent value, which is the most extreme data point that is not an outlier."is not clear
The panel  G "property losses" seems to be inconsistent with the caption (in it there are no  boxes, no whiskers 9 and  an odd number of horizontal bars)
It is not clear which and whether differences among TC characteristics with reference to  track patterns are statistically significant

figure 3
 what is plotted here? the mean, the value of the centroid of each cluster? Infulence (typo), labels on panels are not used (should be deleted)
Improve the caption to explain the meaning of panels and annotations

Figure 4
The unit used "billion ($10^{-4}$%)"  is not clear

Figure 5 the meaning of abbreviations used is not explained in the caption. May be  replace  "damaged" and "undamaged"  with "damage"  and "no damage"? or specify in the caption what  is damaged or "undamaged"

---

## Author Comment (AC1) · 24 Apr 2018

**Response to Referee #1:**

**We would like to thank the reviewer for the time of reading this manuscript and giving suggestions and inputs. Here is our response to the reviewer. The reviewer's comments are in blue ink, and our response follows in black ink. When applicable, the changes made in the manuscript are inserted in italic.**

The article entitled "Track-dependency of tropical cyclone risk in South Korea" by Nam et al. addresses the issue of including cyclone tracks as the bridging factor between exposure and actual impacts of tropical cyclones. The article introduces the main problematic in the first section explaining the distinction between "active" hazards (e.g. rainfall and wind surges) and "potential" hazards (tropical cyclone intensity). Then, an extensive presentation of the datasets, tools and methods that the authors used is followed by a discussion of the results after applying a tree decision method that the authors propose for evaluating cyclones hazard. The method introduced in this paper seems to provide meaningful results when applied to the Korean peninsula and I would recommend the article to be published in NHESS.

This is a good way of summarizing our main idea of track information in risk assessment in the paper, and your positive comment is much appreciated!

My only major concern goes on the presentation and organization of the manuscript. In several parts, I found that the text is repetitive on the arguments and methods and thus less attractive to the reader. Furthermore, my opinion is that the results section is too long and difficult to follow. It should be divided to subsections in order to ease the reader with better articulation of the main findings. Finally, although understandable language could be improved.

It is true that we have endeavored to describe the details in our damage dataset and methods. We have revised the entire manuscript to be more concise and straightforward. Especially method section is rearranged and some parts are simplified. Also, following your suggestion, we divided the result section into three subsections: 3.1 TC hazards and risk of different track types, 3.2 Geographical impacts on TC risk distribution, and 3.3 Decision tree analysis results.

I would also recommend to the authors to include a discussion on the potential application of their method to different regions. It seems that the number of track clusters is detrimental for the complexity of the decision tree. Is the applicability of their method jeopardized if e.g. a broader region with complex geographical and exposure issues is taken into consideration?

First, to answer your question, it is true that for different regions, the number of track clusters can be larger and it can add complexity to the decision tree. However, there are various ways to incorporate track information into decision trees, because decision trees

can deal with both of numerical and categorical data. There are various ways to incorporate track information into decision trees. We adopted clustering method and made it categorical, but for other regions, track information could be divided into two variables such as the approaching angle and distance from the coast.

We have already included discussion of other methods of risk analysis for the potential application to different regions in the discussion section. However, we did not include how to apply the same method of combining fuzzy c-means clustering method (FCM) and decision tree model, for it is very detailed. Instead, we elaborated the reason why we adopted FCM in more details, so that the authors can have better ideas of dealing with track information in their research, if they want. The revised paragraph in the method section is as below:

*"The 85 selected influential TCs are then grouped according to their track patterns using the fuzzy c-means clustering method (FCM). We clustered the track patterns, considering only the part of the tracks in the domain of 28°N–40°N and 120°E–138°E (grey boxes in Fig. 2) so that we could divide tracks focusing on the paths near South Korea, whose national TC risk distribution was examined with respect to these clustered track patterns. Best-track data have 6 hourly longitude and latitude location information of the TCs, and there can be other ways to preprocess track information with the purpose of categorizing their pattern in the area of interest. For example, one can group them with a certain longitude criterion (e.g. east versus west from 128E) or the approaching angle criterion (Hall and Sobel, 2013). We chose to use the FCM, for it is widely used for objectively dividing widespread data with amorphous boundaries. Some previous studies have shown this method to be effective for grouping TC track patterns (e.g., Kim et al. 2011)."*

Finally, I would recommend a demonstration of their method application for the ensemble members of a tropical cyclone forecast. What is the hazard variability when the spread of the forecasted tracks is relatively large?

We introduced decision tree method primarily to assess the relationship of tropical cyclone damage with risk elements. In order to apply these results for prediction, we recommend using random (decision) forest method, which is, in a nutshell, a collection of decision trees. In fact, we are currently studying how to forecast risk with the forecasted tracks and the following hazard forecasts with random forests model. We included one paragraph at the end of the discussion section in our manuscript as below.

*"Finally, if one wants to apply the decision tree methods in other risk analysis, note that decision trees have advantages to other data mining methods that it's easy to interpret, generating straightforward visualizations, but it is prone to overfitting and errors due to bias and variance. It's because decision tree determines an optimal choice at each node. Choosing the best answer at each step does not guarantee the global optimum. We used decision tree method to diagnose the relationship of risk elements in this paper. . If the model makes a different choice at a given step, the final result can be very*

*different, especially when dataset is small. For the current paper, to prevent these errors, we verified our results with pruning and cross-validation. However, for forecasting not understanding, stable and accurate result is more important than transparent and intuitive results. Random forests trains the model with different sample sets of the data (same mechanism to cross-validation in a way). It incorporates random repetition, which makes interpretation and visualization complicated, but random forests give more robust results that a single decision tree."*

---

## Author Comment (AC2) · 24 Apr 2018

**Response to the Editor:**

**We appreciate the editor's time for thorough reading and detailed review of our manuscript. Here we give our response to the comments. The editor's comments are in blue ink, and our response follows in black ink. When applicable, the changes made in the manuscript are inserted in italic.**

This paper shows that for tropical cyclones affecting South Korea, track is the main factor responsible for damages, being more important in a decision tree analysis than other factors such maximum wind speed or minimum atmospheric pressure level.
However, the manuscript needs to be extensively revised for being published. English (though understandable) needs to be improved, many sentences to be rephrased, several paragraph, in particular of the "Results" section to be better focused, figure captions to be more explicative, many typos to be corrected. Further the "Summary and conclusions" section does not deliver the main outcomes of the paper in a clear, direct and concise way. May be better to split it in two sections: "Discussion" and "Conclusions"

We have revised the whole manuscript for better delivery and better presentation, and for improving English. Paragraphs and sentences in many parts are rewritten and rearranged. The result section is divided into three subsections: 3.1 TC hazards and risk of different track types, 3.2 Geographical impacts on TC risk distribution, and 3.3 Decision tree analysis results. Also, "summary and conclusions" section is not splitted into two sections: "Summary" and "Discussion".

Moreover, it should be discussed whether this strong sensitivity of the risk on the tracks is a general characteristic of tropical cyclones or a peculiarity of the South Korea territory, depending on its characteristics in terms of morphology, land use and exposure. Tropical cyclones in the Gulf of Mexico hitting the southern United States would exhibit a similar dependence on their track position?

This is a great question. the sensitive track dependency of TC risk depends on TC-land interaction, so as there are more heterogeneity in geography, there would be more dependency on track. We mentioned Haiyan's case in the Philippines in "Introduction" to indicate that it's not the pattern only appearing in Korea, but we elaborated this point in the introduction including Irma's case, which struck Florida approaching from Gulf of Mexico. The paragraph is as follows:

*"Using TC-based hazard parameters, however, is insufficient for estimating TC damages. Even when TC has a same intensity and size, depending on which track the TC takes and what kind of physical and social geography is along the track, the damage changes drastically. Let's look at the interaction between TC and physical geography first. Record-breaking rainfall in Gangneung city, South Korea was reported, because the track of Typhoon RUSA (2002) was optimal to strengthen the orographic effect on precipitation over the region (Park and Lee 2007). Also, the deadliest damage by typhoon Haiyan in the Philippines in 2013 was mainly because the TC penetrated Tacloban city, which is located in a low-lying area near the ocean,*

*such that most of the damage arose from storm surge (Ching et al. 2015). In both cases, if the TCs went through a different area, avoiding the mountains and lowland, the result could have been much less devastating. Social geography includes the information of the number households are living in the area (exposure) and the building code or preparedness of the community (vulnerability). The role of social geography deciding TC damage is also substantial. Hurricane Irma (2017) deviated ~100 km west from the 5-day forecast of National Hurricane Center, and instead of hitting Miami directly, it struck Tampa where much less urban density is, so Florida avoided the most disastrous scenario."*

The difference between the tracks of the (west and east) cyclone clusters is about 250km. Is this difference sufficiently large to be predictable for an individual cyclone in advance? How many days before reaching South Korea? Authors mention that uncertainty in track of future projections should, therefore, be accounted for. Is this distance among clusters larger or smaller than uncertainty of projections? In there any indication of such a change for South Korea?

The current track forecast accuracy is given in the first paragraph in the result section :
*"This is striking because the short distance around 250 km is somewhat trivial considering that average track errors in the northwest Pacific, as determined by many frequently used dynamic or statistical-dynamical techniques, are about 200 and 400 km for 24 and 48 hours, respectively (Roy and Kovordanyi 2012)."*

The track predictability is similar in Korea that 250 km is in the range of track errors in 48 hour forecast. For future track projection, they use random seeding method, because we cannot predict the genesis location 50-100 years later accurately. So their risk analysis is not a forecast in a strict sense but probabilistic projection. However, even in this probabilistic approach, there is a room to include social, and physical geography and the interaction between TC and the geography before risk materialization in terms of uncertainty.
To answer your question asking if there is a climatic trend in TC track pattern change, there is no significant trend from our time series analysis. However, it is reported from other research that the length of TC track will get longer affecting more mid-latitude countries (Park et al. 2014, and Kossin et al. 2016)

The supplement material contains three tables and a figure to support the description of the decision tree. However, the methodology for construction and validation of the decision tree is not described. I suggest to add a very short text describing it in the main body of the manuscript (in section "Material and methods") and use the supplement for providing more information.

Thank you for this suggestion. We have included cross-validation method and its results in Supplementary following your suggestion. The model should be built in the way to fit best to the training data. In the main result, the training data set and validation data set is the same. However, when the data is changed, when a new TC is approaching, the model can be unstable and error could rise up (over-fitting). Cross-validation method can solve this problem with dividing the training set and validation set (See Reply Fig. 1). Here we present the description of cross-validation and results below. These are newly added in the Supplementary now.

*"Compare to in situ observation based decision tree (Reply Table 2), which has quite stable and accurate results even with cross-validation, Best-track data based decision tree has relatively large error and broad range of distribution in size and accuracy (Reply Table 1). However, 8 of 10 rounds of best-track data based decision tree c-v results show track cluster is the most important attribute to divide damage vs. no-damage cases. The two other rounds picked storm size as the first attribute but they consistently chose track cluster as, at least, the second attribute. The results overall says that the decision tree without in situ observation is not as robust as the one with them, but the significant track-dependency survives through the cross-validation."*

[Figure]

Reply Figure 1. Diagram for 10-fold cross-validation (adopted from Rachka (2015))

Reply Table 1. 10-fold cross-validation results for best-track data based decision tree

| Model | Tree size (mean/std err) | Error rate (mean/std err) | First attribute | Second attribute |
|---|---|---|---|---|
| Original | 8 | **23.4%** | Track cluster | Best-track wind speed & Province |
| 10 rounds of c-v | 9.9 / 0.7 | **31.3% / 2.1%** | | |
| Round 1 | 10 | 40.0% | Track cluster | Province |
| Round 2 | 12 | 31.4% | Best-track radius | Track cluster & Province |

| Round 3 | 8 | 34.3% | Best-track radius | Track cluster & Province |
| Round 4 | 10 | 34.3% | Track cluster | Province |
| Round 5 | 8 | 28.6% | Track cluster | Best-track wind speed & Province |
| Round 6 | 8 | 36.1% | Track cluster | Province |
| Round 7 | 10 | 27.8% | Track cluster | Best-track wind speed & Province |
| Round 8 | 10 | 30.6% | Track cluster | Best-track wind speed & Province |
| Round 9 | 12 | 30.6% | Track cluster | Province |
| Round 10 | 6 | 36.1% | Track cluster | Best-track wind speed & Province |

Reply Table 2. 10-fold cross-validation results for in-situ observation based decision tree,

| Model | Tree size (mean/std err) | Error rate (mean/std err) | First attribute | Second attribute |
|---|---|---|---|---|
| Original | 5 / 0.0 | **12.1% / 0.0%** | Rainfall | Surface Wind |
| 10 rounds of c-v | 5 / 0.0 | **14.4% / 1.5%** | Rainfall | Surface Wind |

Here is a list of minor comments (which is no way meant to be exhaustive)

Thank you again for pointing out typos and suggesting better ways to express the sentences. In general, we have modified the manuscript according to your comments. We would not response line to line for the minor comments, but insert the paraphrased sentences or answer the questions when needed.

Line 6, delete "of"
Line 7 Rephrase the sentence.
Line8 comma missing before "while". Delete "mainly"

Line 10 I suggest "to predict damage." or "to predict the occurrence of damage."
Insert "≤ 250 km" among brackets
Line 12 I suggest to replace "the other hazards developing from a potential to an active hazard" with "an hazard developing from a potential to an active one" …. But may be I do not understand the sentence. Please rephrase.
Line 14-15 "physical geography experience, duration of influence, and relative position of dangerous semicircle side of the TC", this phrasing looks strange to me
Line 16, add comma after "modeling"
 "risk modeling" or "risk assessment"? What is meant with "trivial"?
Line 16 authors, apparently , in the final sentence of the abstract consider "error " and "uncertainty" as equivalent terms. In general they are not. Please explain better.

Abstract is revised incorporating your comments.

*"**Abstract.** In tropical cyclone (TC) risk assessment, many previous studies have attempted to quantify the relationship between TC damage and its ingredients (the risk elements–exposure, vulnerability, and hazard). For hazard parameters, TC intensity and size information, such as central minimum pressure, maximum wind speed, and 30 knot radius of the TC, have been widely utilized. However, our risk analysis of 85 TCs that made landfall in South Korea during 1979-2010, shows that a small deviation of west-east in TC track (£ 250 km, smaller than the average radius of TC) is more important than TC intensity/size for deciding the amount and distribution of TC damage in South Korea This significant track-dependency of TC damage exists because TC track is responsible for the realization of a hazard developing from a potential to an active one. Locally experienced rainfall and wind-gust hazard are not represented well by the bulk indices of TC intensity/size. Plus, the complexity in terrain and heterogeneity in urban landscapes of South Korea should also contribute to this sensitive track-dependency of TC risk. These results suggest that, when we attempt to assess future TC risk, the role of land-atmosphere interaction should be considered more carefully. Given the large uncertainty of the TC track prediction from current global climate models, the bulk approach may give misleading guidelines for future risk distribution."*

End of section 1: a description the content of the paper is missing

Following sentences are inserted at the end of section 1:
*"The rest of the paper is organized as follows. Section 2 lists the data sets for information of TC, local hazard, damage, and social index, used in this study, and explains how these dataset was processed and statistically analyzed. Results from risk comparison analysis and decision tree analysis are described in section 3. Finally, the results are interpreted and summarized in section 4 and discussions on applications and implication of this research are presented in section 5."*

Section 2 replace "Materials and method" with "Data and methods"

Section name changed.

Page 3 line 1, definition of TC size is unclear

Following sentence is inserted.
"For TC size, we used the longest radius of 30 knot winds, which is specifically provided by RSMC."

Line 4 follwing, consist,
Line 5: "standardized to the value of money in 2005 and taking inflation into account" Should "and" be replace with "by"? otherwise it looks a duplication of the same concept
Line 6 avoid repeating "collected"
Line 8 was including

Manuscript is modified according to above comments.

Line 10 bad phrasing, The damage was likely indeed caused by waves, which in turn were caused by the TC

The sentence was rewritten as following: *"... some cases were categorized under high-wave damage, when in fact it was also a TC damage for the high waves were caused by a TC."*

Line 16 bad phrasing. I suggest replacing "if there exist any" with "the presence of"
Probably "if" should be replaced with "whether"

The sentence was rewritten as following: *""No damage" and "Damage" cases were later categorized based on whether there exist any economic loss records reported by NDIC for the given province for the given TC event."*

Line 14 what is the "damage period"?

We recognize the wording of damage period is confusing. We added a footnote as follows: *"[1] NDIC cannot differentiate the damage from multiple hazards when there are multiple successive extreme phenomena. For example, if heavy rainfall watch started on July 15th and then a TC came to South Korea on July 20th and decayed on July 22th, and there was no gap between the rainfall and TC advisories, NDIC aggregates the damage amounts and record the damage period as July 15th to 22th. Therefore, to confine the origin of the loss data to one TC, we excluded cases whose damage period exceed five days from landfall."*

Line 25 delete "of"
Line 26 delete "of provinces"

Changes are made in the manuscript.

Line 29 what is the "influence duration"? delete "also"
Line 31 rephrase "the range of duration was limited by the summation". Sentence is not clear

In the revised manuscript, we mention how we defined influence duration from weather stations in result section, when we are actually describing the influence duration because we found it is more effective to explain there.

*"Here we highlight the importance of rainfall hazard from TC again by pointing out that the spatial patterns of influence duration and precipitation are almost coincident (compare 2ⁿᵈ row and 3ʳᵈ row of Fig. 4. We calculated influence duration for each station by applying the same criteria for wind and rainfall. A station is marked as "influenced" if either of daily accumulated precipitation or daily maximum sustained wind speed recorded at that station at the specific day exceeded the station's critical thresholds, which we set as the 90ᵗʰ percentile of each station. Here, influence duration was largely determined by the criteria of rainfall not wind echoing the correlation analysis result above (Table 1)"*

Line 1 "defined and distinguished" better "identified"?

Changes are made in the manuscript.

Line 4-5 "influential" means having great influence on someone or something. This criterion does not account in any way for the size of the impact of the cyclone
Line 14 "more comprehensive" than what?

We simplified the description of this part following the comments from referee #1.

*"Then, we verified these TCs with the official influential TC record in the Typhoon White book issued by the Korean National Typhoon Center (NTC, 2011) as in our previous study (Refer to Park et al. (2016) for more details of NTC Typhoon White book)."*

Line 18 replace "from" with "in". "their intensity was above TS" replace with "their wind speed was above the TS threshold"

Changes are made in the manuscript.

Line 29-31 The explanation of the grouping criteria and why four clusters have been used is not clear to me. What is here a validity measure? The definition of the used indices is missing
Line 26 replace "not for the whole tracks from genesis to disappearance, but for" with "considering only"
Line 26-28 long sentence
I stop here with comments on the English form
Line 30 "Xie and Beni index, and Dunn index" should be briefly described

This section is elaborated to give more information about validity measure as follows:

*"The TCs were grouped into four types. The optimum cluster number were decided by four validity measures - partition coefficient, partition index, separation index (i.e. Xie and Beni*

*index), and Dunn index. The partition coefficient measures how much overlapping the fuzzy clusters, and other three indices measure the degree of compactness and separation of the clusters. Larger partition coefficient and smaller partition index, separation index and Dunn index make better clustering (For more detailed explanation and formula of validity measures for the optimum cluster number, refer to Appendix B of Kim et al. 2011). All of the indexes pointed to four to be the optimum number in our case. We had some sensitivity tests making slight changes to TC lists, such as different time frame (e.g., 1979–2015) or different clustering domain (e.g., 5° area from the Korean Peninsula coastline), and four still appeared to be the optimum cluster number from the validity measures.*"

Line 16-17 what is meant here with stability and consistency of results? How was this checked?

Answered above (Reply Fig. 1, Reply Table 1 and 2).

Section RESULTS
line 1 distance between types? How is defined the distance between two types

It is distance between the mean tracks, and that information is added as follows: *"Although zonal distances between the mean track of east-types (i.e., east-short and east-long) and the mean track of west-types (i.e., west-short and west-long) are only about 250 km, ..."*

Line 11 "TC-based hazard ranking" on which variable is based? How is the ranking computed?
Line 15 … apparently a self-contradicting statement
line 11-12 sentence: ranking based on which quantity?
In my view the second paragraph not well focused. It addresses some differences between hazards rankings among clusters, between this and previous studies, and among hazards …all together . please explain better

Following the comments, we rewrote the paragraph splitting into two paragraphs as below:

*"As expected, TC-based hazards display different results from active hazards. Although TC-based hazard parameters have been commonly used as the sole indicators in TC risk analysis (e.g., Nordhaus 2010; Hsiang and Narita 2012; Czajkowski and Done 2014; Zhai and Jiang 2014), it shows poor accordance with damage, especially when compared to active hazard. For TC intensity and size (TC-based hazard parameters), longer tracks (i.e., east-long and west-long) have larger values, however for active hazards, west-types generally have higher values than east-types (compare Fig. 3(a)-(c) to (d)-(f)). Then damage from the different track types is correlated much more with active hazards rather than TC intensity and size (Table 1).*

*To look at Fig. 3 more closely, for all of maximum wind speed, central pressure, and 30 knot radius, the ranking is in order of east-long, west-long, west-short, and east-short. On the other hand, for near-surface wind, the ranking is in order of west-long, west-short, east-long, and east-short. The damage ranking is in order of west-short, west-long, east-long, and east-short*

*track patterns, which is exactly the same as the ranking for accumulated precipitation and influence duration. Table 1 shows all of the active hazard parameters considered here show much higher correlations with damages than TC-based potential hazard parameters, even if most of TC-based hazards display statistical significance at the 95% or 99% confidence. The average of absolute correlation coefficient (| r |) for all active hazards and for all track patterns is 0.62, while that of potential hazards is just 0.29. One thing we want to point out is that higher correlation coefficients for accumulated precipitation and affected duration compared to near-surface wind imply that rainfall is the main cause for damage from TCs in the study area rather than wind, as suggested by some previous studies (Lin et al. 2002; Park et al. 2016)."*

I think that "damaged " or "undamaged" can refer to the territorial units, to the population, to the exposed goods, but not to a "case". The expression "damaged case" sounds odd to me

We have changed the terms to "damage" or "no damage" throughout the paper.

line 2 "navigable"?

Navigable hemicircle and dangerous hemicircle is west-side and east-side of TC in northern hemisphere, it is an estabilished term in TC studies but we deleted this term as it is not essentially needed to explain our points.

Third paragraph long and not well focused
Line 29-30 repetition

We have rewritten the paragraphs dividing them according to different main ideas.

Section CONCLUSION
In the text I cannot find a precise definition of Active hazard (it should be provided). Anyway, my understanding is that they are hazard that actually produce some losses, victims, accidents emergency. In such case the sentence at line 5-6 page 9 is trivial as it follows for the definition of active hazard .Pease explain better or deleted it.

We have largely reorganized and rewritten the "Summary" and "Discussion" part. For the definition of active hazards we added a paragraph in introduction to clarify the terminology. We also moved the diagram in Figure 6 to Figure 1 to present the framework from the first.

Following paragraph is inserted in introduction:
*"In this paper, we adopt **indirect** versus **direct cause** concept from causality science (Fig. 1 of Ebert-Uphorff and Deng 2012) in association with the hazard mode concept (**potential** versus **active hazard**) from risk management field (MacCollum 2011), and we cooperate them into risk triangle framework. For hazard mode concept, active hazard refers to "a harmful incident involving the hazard has actually occurred", whereas potential hazard refers to the situation environment is currently affected but there not yet activated at a given place and at a given time (MacCollum 2011). By this definition, we refer heavy rainfall, wind gust, or surge induced at the*

*local area by the TC to active hazards, and consider approaching TC system as a potential hazard. In causality relationship then, TC intensity or size variables are indirect causes for damage occurrence and local active hazards are direct cause for damage. The cause–effect relationship between a TC and damage occurrence at a settlement always goes through the variable of local active hazards (See Fig. 1 for the diagram). In other words, if we want to make a prediction for whether there would be damage or not at a city, and we already know the local hazards information of precipitation, wind speed, and surge heights there, we do not gain any additional information by knowing the central intensity of the TC system. Figure 1 shows the graphical model summarizing above points, and indicating the position of track in causality relationship of TC risk process that we propose in this paper."*

Following paragraph is inserted in summary:
*"We recall Fig. 1 as a concluding remark. This framework includes track as a bridging component between TC hazard (indirect cause) and local active hazard (direct cause), because TC hazard can be activated only through track. In other words, only through the "conflicts at the interface between geophysical processes and human societies" (Alexander 2000). Then, the risk triangle is applied not to the potential hazard (TC intensity and size) but to active hazard, which is a product of a combination of TC characteristics (i.e. dormant hazards) and local geography experiences through track. Note that not only local geography experience is dependent on track patterns, but TC characteristics also appeared to differ among track patterns (Figs. 3(a) - (c))."*

Line 11 "When local active hazard information is missing, TC track acts to bridge the information gap between the TC system and local risk" is not clear to me

This sentence came from the interpretation of Supplementary Fig. 1 (in-situ observation based decision tree). However, we did not explain the Supplementary Fig. 1 enough to make this conclusion. It felt like to much digression to include lengthy description for Supplementary Fig. 1, so we deleted this sentence.

FIGURES
figure 2
first and third quantile values are 0.25 and 0.75? please be specific
The sentence "The plotted whiskers extend to the adjacent value, which is the most extreme data point that is not an outlier."is not clear
The panel G "property losses" seems to be inconsistent with the caption (in it there are no boxes, no whiskers 9 and an odd number of horizontal bars)
It is not clear which and whether differences among TC characteristics with reference to track patterns are statistically significant

figure 3
what is plotted here? the mean, the value of the centroid of each cluster? Infulence (typo), labels

on panels are not used (should be deleted)
Improve the caption to explain the meaning of panels and annotations

Figure 4
The unit used "billion (10-4%)" is not clear

Figure 5
 the meaning of abbreviations used is not explained in the caption. May be replace "damaged" and "undamaged" with "damage" and "no damage"? or specify in the caption what is damaged or "undamaged"

We have modified all of our figures following your comments, and also we changed the labels from upper class (A, B, C) to lower class in the parentheses as requested ((a), (b), and (c)) in the guideline of NHESS.

**Reference:**

Roy, C., and Kovordanyi, R.: Tropical cyclone track forecasting techniques–A review, Atmos. Res. 104-105, 40-69, doi: 10.1016/j.atmosres.2011.09.012, 2012.

Park, D.-S. R., Ho, C.-H., and Kim, J.-H.: Growing threat of intense tropical cyclones to East Asia over the period 1977-2010. Environ. Res. Lett., 9, Artn 014008, 2014.

Kossin, J, Emanuel, K. A., and Camargo, S. J.: Past and projected changes in western North Pacific tropical cyclone exposure, J. Climate, 29, 5725-5739, doi:10.1175/JCLID-16-0076.1, 2016.

Raschka, S: Python Machine Learning. Packt Publishing, 454 pp. 2015

---

## Author Response (AR1)

We would like to thank the reviewer and the editor for the time of reading this manuscript and giving suggestions and inputs. Here is our response. The comments of reviewer and editor are in black ink, and our response follows in bold and blue ink. When applicable, the changes made in the manuscript are inserted in italic.

**Response to Referee #1:**

The article entitled "Track-dependency of tropical cyclone risk in South Korea" by Nam et al. addresses the issue of including cyclone tracks as the bridging factor between exposure and actual impacts of tropical cyclones. The article introduces the main problematic in the first section explaining the distinction between "active" hazards (e.g., rainfall and wind surges) and "potential" hazards (tropical cyclone intensity). Then, an extensive presentation of the datasets, tools and methods that the authors used is followed by a discussion of the results after applying a tree decision method that the authors propose for evaluating cyclones hazard. The method introduced in this paper seems to provide meaningful results when applied to the Korean peninsula and I would recommend the article to be published in NHESS.

My only major concern goes on the presentation and organization of the manuscript. In several parts, I found that the text is repetitive on the arguments and methods and thus less attractive to the reader. Furthermore, my opinion is that the results section is too long and difficult to follow. It should be divided to subsections in order to ease the reader with better articulation of the main findings. Finally, although understandable language could be improved.

Reply) Thank you for your comments. We have revised the entire manuscript to clearly suggest the importance of track in TC risk analysis. We divided the result section into two subsections: 3.1 TC hazards and risk of different track types & 3.2 Importance of track in TC risk analysis following the reviewer's suggestion. To improve English expression, we also got the English grammar correction service before the submission.

I would also recommend to the authors to include a discussion on the potential application of their method to different regions. It seems that the number of track clusters is detrimental for the complexity of the decision tree. Is the applicability of their method jeopardized if e.g. a broader region with complex geographical and exposure issues is taken into consideration?

Reply) Thank you for excellent idea! Although we are afraid that this is out of scope of our study, we indeed agree to the reviewer's suggestion. Hence, we just made some discussion on possible difference in importance of track in TC risk between countries as follows.

"On the other hand, the importance of track may differ by country because topography among the three factors suggested is not identical. If a country has major mountainous area like South Korea, track information may become more important, and vice versa. The dependence of track in TC risk over Southeastern United States, for example, in which there is little mountainous area, may be less important than that of South Korea. As a future study, we would compare role of track in TC risk between countries having different topographic conditions."

**Response to the Editor:**

This paper shows that for tropical cyclones affecting South Korea, track is the main factor responsible for damages, being more important in a decision tree analysis than other factors such maximum wind speed or minimum atmospheric pressure level. However, the manuscript needs to be extensively revised for being published. English (though understandable) needs to be improved, many sentences to be rephrased, several paragraph, in particular of the "Results" section to be better focused, figure captions to be more explicative, many typos to be corrected. Further the "Summary and conclusions" section does not deliver the main outcomes of the paper in a clear, direct and concise way. May be better to split it in two sections: "Discussion" and "Conclusions"

Reply) Thank you for this comment. We have revised the entire manuscript to clearly suggest the importance of track in TC risk analysis. We made summary and conclusions section much more simplified to better deliver our main idea instead of splitting it into two sections. To improve English expression, we also got the English grammar correction service before the submission.

Moreover, it should be discussed whether this strong sensitivity of the risk on the tracks is a general characteristic of tropical cyclones or a peculiarity of the South Korea territory, depending on its characteristics in terms of morphology, land use and exposure. Tropical cyclones in the Gulf of Mexico hitting the southern United States would exhibit a similar dependence on their track position?

Reply) As we replied to the reviewer #1's comment, this is an excellent idea to discuss whether the strong sensitivity of TC risk on track may vary by regions. While we cannot explicitly suggest if the southern US would exhibit a similar dependence on tracks or not, but it is likely different from South Korea case. This is because there is little mountainous area in southern US than South Korea. We made some discussion on possible difference in importance of track in TC risk between countries as follows.

"On the other hand, the importance of track may differ by country because topography among the three factors suggested is not identical. If a country has major mountainous area like South Korea, track information may become more important, and vice versa. The dependence of track in TC risk over Southeastern United States, for example, in which there is little mountainous area, may be less important than that of South Korea. As a future study, we would compare role of track in TC risk between countries having different topographic conditions."

The difference between the tracks of the (west and east) cyclone clusters is about 250km. Is this difference sufficiently large to be predictable for an individual cyclone in advance? How many days before reaching South Korea? Authors mention that uncertainty in track of future

projections should, therefore, be accounted for. Is this distance among clusters larger or smaller than uncertainty of projections? In there any indication of such a change for South Korea?

Reply) The 250-km distance is not long considering that average errors of track forecasting in the western North Pacific are about 200 and 400 km for 24 and 48 hours, respectively (Roy and Kovordanyi 2012). Hence, this high sensitivity of damage on track shown in Fig. 3 suggests that the current skill of TC track forecasting may not be enough to exactly estimate TC risk distribution over South Korea in advance of 1 day and over. We added the related explanation in the revised manuscript as follows.

"Although the average zonal distance between the mean tracks of east-types (i.e. east-short and east-long) and west-types (i.e. west-short and west-long) was only about 250 km, hazards (both potential and active) and damages caused by the TCs are significantly different depending on the four TC track patterns at the 99% confidence level based on the Kruskal-Wallis test (Fig. 3). This highlights the importance of track in TC risk assessment because the 250 km distance is not long considering that the average errors of track forecasting in the western North Pacific are about 200 and 400 km for 24 and 48 hours, respectively (Roy and Kovordanyi 2012). Meanwhile, the high sensitivity of damage on the track shown in Fig. 3 suggests that the current skill of TC track forecasting may not be enough to exactly estimate TC risk distribution over South Korea in advance of 1 day and over."

In addition, we deleted the sentence on future uncertainty in the abstract because we did not explicitly look into that in the present study. Instead, we just discussed on it in the conclusion and discussion section as follows.

"Our results also suggest that it is necessary to consider possible large uncertainty in future TC risk projection because of high sensitivity of TC risk on track, as well as the lack of reliability of future projection of TC tracks (Knutson et al. 2010, Walsh et al. 2015)."

The supplement material contains three tables and a figure to support the description of the decision tree. However, the methodology for construction and validation of the decision tree is not described. I suggest to add a very short text describing it in the main body of the manuscript (in section "Material and methods") and use the supplement for providing more information.

**Reply**) We have included the description on cross-validation method in the revised manuscript, and have provided the associated supplementary table (See the last paragraph of section 2.2.1 Data mining methods)

Here is a list of minor comments (which is no way meant to be exhaustive)

Reply) Thank you for the editor's helpful comments. We are sorry for not giving lineby-line responses, however, we have extensively revised manuscript including figure captions following the comments. We marked the changes associated with important reviewer comments with sky blue ink in the revised manuscript.

Page 7 line 2 "navigable"?

Reply) Navigable circle is a terminology used in meteorology. For readers who are unfamiliar with the term, we added explanation on it in the revised manuscript as follows.

"This can be attributed to the concepts of dangerous and navigable semicircles. In the case of west-type tracks, South Korea falls within a dangerous semicircle (righthand side of the direction of TC movement), in which the TC translation speed and rotational wind field are additive, and hence strong wind speed is observed therein. In contrast, in the case of east-type tracks, the country is located under a navigable semicircle (left-hand side of the direction of TC movement), in which the TC translation is counter-directional to the rotational wind. Therefore, weaker wind speeds are found there than that in the dangerous semicircle."

Line 30 "Xie and Beni index, and Dunn index" should be briefly described

**Reply) We briefly described them in the revised manuscript as follows.**

"The partition coefficient measures how much overlapping the fuzzy clusters have, and inversely proportional to the average overlap between the clusters. Both of the partition and separation indices are computed by compactness and separation of the clusters. However, the partition index represents separation as the sum of the distances between the clusters while the separation index does as the minimum of them. The Dunn index is calculated by the ratio of the shortest and the longest distances of the two objectives within a same cluster. The larger partition coefficient and smaller partition index, separation index, and Dunn index create better clustering (for a more detailed explanation and formula of validity measures for the optimum cluster number, refer to Appendix B of Kim et al. 2011)."

**Section CONCLUSION**

In the text I cannot find a precise definition of Active hazard (it should be provided). Anyway, my understanding is that they are hazard that actually produce some losses, victims, accidents emergency. In such case the sentence at line 5-6 page 9 is trivial as it follows for the definition of active hazard .Pease explain better or deleted it.

**Reply**) We now define the different modes of hazard more explicitly at the first part of result section in the revised manuscript as follows.

"In this paper, we adopted the hazard mode concept (**potential** versus **active hazard**) from the risk management field (MacCollum 2006). For the hazard mode concept, active hazard refers to a situation when "a harmful incident involving the hazard has actually occurred", whereas potential hazard refers to a situation where "the environment is currently affected but not yet activated at a given place and time". By this definition, we refer to heavy rainfall and wind gust induced at the local area by the TC as active hazards, and we consider the TC system's minimum central pressure, maximum wind speed, and size over South Korea as potential hazards. These two modes of TC hazard (potential and active) are utilised throughout this paper."

Line 6, delete "of"

Line 7 Rephrase the sentence.

Line8 comma missing before "while". Delete "mainly"

Line 10 I suggest "to predict damage." or "to predict the occurrence of damage."

Insert "≤ 250 km" among brackets

Line 12 I suggest to replace "the other hazards developing from a potential to an active hazard" with "an hazard developing from a potential to an active one" …. But may be I do not understand the sentence. Please rephrase.

Line 14-15 "physical geography experience, duration of influence, and relative position of dangerous semicircle side of the TC", this phrasing looks strange to me

Line 16, add comma after "modeling"

"risk modeling" or "risk assessment"?

What is meant with "trivial"?

Line 16 authors, apparently, in the final sentence of the abstract consider "error" and "uncertainty" as equivalent terms. In general they are not. Please explain better.

End of section 1: a description the content of the paper is missing

Section 2 replace "Materials and method" with "Data and methods"

Page 3 line 1, definition of TC size is unclear

Line 4 follwing, consist,

Line 5: "standardized to the value of money in 2005 and taking inflation into account" Should"and" be replace with "by"? otherwise it looks a duplication of the same concept

Line 6 avoid repeating "collected" Line 8 was including

Line 10 bad phrasing, The damage was likely indeed caused by waves, which in turn were caused by the TC

Line 16 bad phrasing. I suggest replacing "if there exist any" with "the presence of" Probably "if" should be replaced with "whether"

Line 14 what is the "damage period"?

Line 25 delete "of" Line 26 delete "of provinces"

Line 29 what is the "influence duration"? delete "also" Line 31 rephrase "the range of duration was limited by the summation". Sentence is not clear

Line 1 "defined and distinguished" better "identified"?

Line 4-5 "influential" means having great influence on someone or something. This criterion does not account in any way for the size of the impact of the cyclone Line 14 "more comprehensive" than what?

Line 18 replace "from" with "in". "their intensity was above TS" replace with "their wind speed

was above the TS threshold"

Line 29-31 The explanation of the grouping criteria and why four clusters have been used is not

clear to me. What is here a validity measure? The definition of the used indices is missing Line 26 replace "not for the whole tracks from genesis to disappearance, but for" with "considering only"

Line 26-28 long sentence

I stop here with comments on the English form

Line 16-17 what is meant here with stability and consistency of results? How was this checked?

Section RESULTS Page 6 line 1 distance between types? How is defined the distance between two types

Line 11 "TC-based hazard ranking" on which variable is based? How is the ranking

**computed?**

Line 15 ... apparently a self-contradicting statement line 11-12 sentence: ranking based on which quantity? In my view the second paragraph not well focused. It addresses some differences between hazards rankings among clusters, between this and previous studies, and among hazards ...all together . please explain better

I think that "damaged " or "undamaged" can refer to the territorial units, to the population, to the

exposed goods, but not to a "case". The expression "damaged case" sounds odd to me

Third paragraph long and not well focused Line 29-30 repetition

Section CONCLUSION

Line 11 "When local active hazard information is missing, TC track acts to bridge the information gap between the TC system and local risk" is not clear to me

**FIGURES**

figure 2

first and third quantile values are 0.25 and 0.75? please be specific

The sentence "The plotted whiskers extend to the adjacent value, which is the most extreme data point that is not an outlier." is not clear

The panel G "property losses" seems to be inconsistent with the caption (in it there are no boxes, no whiskers 9 and an odd number of horizontal bars)

It is not clear which and whether differences among TC characteristics with reference to track patterns are statistically significant

figure 3

what is plotted here? the mean, the value of the centroid of each cluster? Infulence (typo), labels

on panels are not used (should be deleted)

Improve the caption to explain the meaning of panels and annotations

Figure 4

The unit used "billion (10-4%)" is not clear

Figure 5

the meaning of abbreviations used is not explained in the caption. May be replace "damaged" and "undamaged" with "damage" and "no damage"? or specify in the caption what is damaged or "undamaged"